# Electrochemical Machining of Curvilinear Surfaces of Revolution: Analysis, Modelling, and Process Control

**DOI:** 10.3390/ma15217751

**Published:** 2022-11-03

**Authors:** Jerzy Sawicki, Tomasz Paczkowski

**Affiliations:** 1The Mechanics and Computer Methods Department, Bydgoszcz University of Science and Technology, Al. Prof. S. Kaliskiego 7, 85-796 Bydgoszcz, Poland; 2The Manufacturing Techniques Department, Bydgoszcz University of Science and Technology, Al. Prof. S. Kaliskiego 7, 85-796 Bydgoszcz, Poland

**Keywords:** electrochemical machining, electrolyte flow, mathematical modelling, computer simulation, method of perturbation, adaptive control process

## Abstract

The paper presents the authors’ model for the adaptive control of the electrochemical machining (ECM) process of machining the rotary (axisymmetric) elements of any curvilinear shape, using the results of theoretical computer simulation of this process. Computer simulations have been based on the authors’ model of the ECM of rotary surfaces of any curvilinear shape. The quasi- 3D ECM model proposed facilitates an analysis of physical phenomena which occur in the interelectrode gap. Mathematical ECM modelling has been based on the application of the equation of the workpiece shape evolution and on the system of partial differential equations resulting from the principle of mass conservation, momentum and the law of conservation of energy describing a flow of the mixture of electrolyte in the interelectrode gap. A solution to the problem has been developed with analytical and numerical integration. For the rotary hemispheric surface, in a set time, the local machining of a change in the interelectrode gap thickness and characteristic physicochemical parameters were determined, especially static pressure distribution, electrolyte flow velocity, temperature and volumetric gas phase concentration as well as current density. The simulation results were experimentally verified by determining the distribution of the shape deviation (WP) calculated from the process computer simulation and after the ECM. Applying the adaptive control of the ECM process has facilitated, based on the simulations made, enhancing the process stability and avoiding the occurrence of critical states.

## 1. Introduction

Development of technology results in the industry applying construction materials harder and harder as well as stronger and stronger, with resistance to corrosion, to the effects of high temperatures and other factors. Most of those materials are difficult to machine with traditional methods.

The electrochemical machining (ECM) patented in 1928 [1,2], introduced to the aviation industry after WWII, solves many manufacturing problems effectively. The key ECM advantage is the possibility of manufacturing pieces complex in shape made of materials hard to machine without tool wear [3,4]. Despite the high costs of specialist machine tools, power supply units, adequate electrodes [5,6,7,8], controls, and problems with the selection of the right values of the machining parameters, ECM has been still developing intensively [9,10,11,12]. 

The basic ECM method is the use of a contouring electrode with a shaping tool electrode. Applying special electrodes, additional movements of the workpiece (WP) and a working electrode (TE), it is possible to perform electrochemical turning, electrochemical milling (vibrating electrode machining) [13,14,15,16,17], machining with complex kinematics corresponding to copy milling with a universal electrode [18,19,20,21,22] as well as hybrid machining: electrochemical-abrasive (grinding, honing) and electrochemical-electrical discharge [23,24]. In all those machining methods, supplied with a direct current, pulsating direct current or pulsed current is possible. The feed motion frequently overlaps with an additional oscillating motion with a set amplitude and frequency with the possibility of synchronizing the motion with the induced pulses of working voltage.

Due to the costs, the complexity of the ECM process, and the development of competitive discharge methods, electrochemical machining is currently applied in the aerospace, aviation, arms, and electronic sectors. Many global companies apply it in the processes of manufacturing the rotor blades and guide vanes of compressors and turbines [7,11,25,26], cooling holes in turbine blades [27], complex moulds, and respective responsible parts made of special alloys [28,29]. The ECM is also applied for deburring [30,31], smoothing the surfaces [32,33], especially in hardly accessible places [24,28,34,35].

The experience acquired throughout many years of applying ECM has demonstrated that the process poses many problems which do not occur in traditional treatments. The problems are related to the internal structure of the ECM for high current densities [9,10,36,37,38,39]. It concerns electrode processes, the exchange of mass, momentum, and energy, electric charge transfer, flow hydrodynamics as well as variable properties of the medium in the interelectrode gap (IEG) [3,4,40,41,42,43]. 

The multidimensional and dynamic ECM process requires applying the computer system of machining design and control [44,45,46,47].

The aspect of the ECM process covers [9,10,48,49,50]:the analysis of a change in the workpiece (WP) shape in time,determining the final shape of the workpiece (WP),determining the geometry of the working electrode to provide the desired shape of the workpiece (WP),optimising the process conditions to minimise the workpiece (WP) shape errors,searching for new methods to enhance machining accuracy.

To solve those problems, a comprehensive analysis of physical phenomena which occur in the process of electrochemical surface development is required [51,52,53,54,55]. A considerable number of parameters controlling the ECM process, disturbing the process, as well as very frequently interdependent, make it difficult to formulate a general ECM theory, considering the real physical conditions of the process [56,57].

Abundant ECM literature seems not to provide a comprehensive coverage of the mathematical modelling of this process for any curvilinear surfaces of revolution considering the aspects of adaptive control.

The papers found in literature discussing the electrochemical machining of the surfaces of revolution concern mostly the machining of frontal and lateral surfaces of rotary workpieces, electrochemical grinding, and treatment which involves deburring, calibrating holes, and smoothing the rotary surfaces [33,58,59,60,61].

Some aspects of the modeling of the electrochemical treatment of rotary-shaped surfaces are covered by Kozak and Dąbrowski [9,10]. An article by Dąbrowski [9] provides an analysis of the ECM of a fixed axisymmetric shape surface based on an approximate one-dimensional flow model. The analysis provides static pressure distributions, mean velocity, temperature, gas phase concentration, and interelectrode (SM) gap thickness. The paper [10] signals a problem with the ECM of the fixed axisymmetric surface from the ideal ECM model and a simplified two-dimensional electrolyte flow model. 

This paper presents the authors’ adaptive control model for the ECM process of rotary (axisymmetric) workpieces. This model uses the results of computer simulations for arbitrary revolving curvilinear surfaces. The quasi-3D machining model proposed facilitates an analysis of physical phenomena which occur in the interelectrode gap. The analysis and mathematical ECM modelling are based on the use of the WP surface shape evolution equation and a system of partial differential equations resulting from the principles of mass, momentum, and energy conservation describing an electrolyte flow in the interelectrode gap. The solution to the system was made with analytical and numerical integration. 

The aim of mathematical modelling of ECM curvilinear rotary surface machining is to determine the local changes in the interelectrode gap thickness following machining at a set time as well as characteristic physicochemical parameters, especially static pressure distribution, electrolyte flow velocity, temperature, and volumetric gas phase concentration, and current density. 

The application of adaptive control of the ECM process will enhance, based on the simulations, the process stability and help to avoid the occurrence of critical states.

## 2. Materials and Methods

### 2.1. Scheme of Electrochemical Machining Process Modeling

The scheme of ECM process for curvilinear surfaces of revolution is shown in Figure 1.

The treatment process is carried out by anodic dissolution. The workpiece (WP) is a positive electrode (anode) and the tool (TE) is a negative electrode (cathode). The space between the anode and the cathode is filled with electrolytes. The flow of current between the electrodes causes anodic dissolution, which leads to the removal of the material from the anode (workpiece).

The WP (anode) rotates at set speed n and the working electrode TE (cathode) makes a feed motion along the Z axis at the set speed $v_{f}=$ *const*. The IEG is filled with an electrolyte the forced flow of which is constant flow rate *Q = const*. 

The electrode area is described with function *R*(*x*) which is the radius of the area. The thickness of IEG *h* is the section made along the normal to the TE area.

### 2.2. Theoretical Analysis of ECM Process

The mathematical model of the ECM process for rotating surfaces is based on the following basic assumptions:the flow of the mixture (electrolyte, hydrogen) is two-phase, homogeneous, non-slip,the distribution of the gas phase results from the intensity of the treatment process and is determined by the volume concentration of hydrogen *β*(*x*),the phase of the digestion products of the anode is omitted,the anode oxygen phase is omitted (the current efficiency of oxygen is assumed to be zero),the temperature and pressure of the gas in the bubbles are equal to the temperature and pressure of its surroundings,the value of the electrochemical digestion coefficient *k_V_* is determined on the basis of experimental tests of the anode-electrolyte-cathode set,to determine the hydrodynamic parameters, a stationary (steady), two-dimensional (axisymmetric) flow of the electrolyte and hydrogen mixture is assumed.

Other assumptions required to model the ECM treatment process are presented below.

Gap thickness *h(x)* is low as compared with the electrode area radius *R*(*x*). Hence: (1)h(x)<<R(x)

The equation of the WP surface shape evolution as a result of anode dissolution assumes the form of [8,9,10,62,63,64]:(2)$$\frac{\partial Z_{A}}{\partial t}=k_{V}j_{A}\sqrt{1+{\left(\frac{\partial Z_{A}}{\partial R}\right)}^{2}}\;$$
for t=0 $Z_{A}=Z_{A}\left(R\right)$.

where: $k_{V}$—is the coefficient of electrochemical machinability which is defined as the volume of material dissolved per unit electrical charge, $Z_{A}\left(R\right)$—describes an initial shape of the WP surface.

Assuming the linear potential distribution along the normal segment (often assumed in technological calculations) to the TE, the current density function can be expressed with the following dependence [8,10,26,63]:(3)$$j_{A}=\kappa_{0}\Phi_{TG}^{-1}\frac{U-E}{h\left(x\right)}\;$$
where: $\kappa_{0}$—electrolyte conductivity at $T_{0}$ and β=0, $T_{0}—$initial temperature of the electrolyte, *h*—the lowest IEG thickness, *β*—void fraction (volumetric gas concentration), U—working voltage between the cathode TE and the anode WP, *E*—total over potential at the gap inlet.

The lowest IEG thickness is the distance of the point on the WP anode from the point on the TE surface described with the following dependence:(4)$$Z_{e}\left(R,t\right)=f\left(R_{e}\right)+v_{f}t\;$$
where: $f\left(R_{e}\right)—$coordinate of TE, $v_{f}$—feed rate of the TE.

Function *Φ_TG_* describes changes in conductivity in the IEG triggered by the variable field of temperature and volumetric gas phase concentration and it is determined as follows [9,10,65]: (5)$$\Phi_{TG}=\frac{1}{h}\left[\int_{0}^{h}\frac{dy}{\left(1+\alpha_{T}∆T\right){\left(1-\beta \right)}^{\frac{3}{2}}}\right]\;$$
where: $\alpha_{T}$—conductivity coefficient of the electrolyte at *T*, $∆T=T-T_{0}—$temperature increment.

To determine the WP shape evolution (solution of Equation (2), it is necessary to determine temperature increments ∆T=T−To and distribution void fraction β.

It requires solving a system of equations for electrolyte movement in the IEG. 

The mathematical modelling of the flow of electrolytes in the IEG was made in the locally orthogonal system of coordinates related to the TE x,y [65].

Treating the electrolyte as a mixture of electrolyte and hydrogen, the equations of motion resulting from the principles of mass, momentum, and energy conservation in the locally orthogonal coordinates system assuming a laminar flow assume the following form [1,8,64]:-flow continuity equations for electrolyte and hydrogen, respectively:
(6)$$\frac{1}{R}\frac{\partial \left(\rho_{e}Rv_{x}\right)}{\partial x}+\frac{\partial \left(\rho_{e}v_{y}\right)}{\partial y}=0\;$$
(7)$$\frac{1}{R}\frac{\partial \left(\rho_{h}Rv_{x}\right)}{\partial x}+\frac{\partial \left(\rho_{h}v_{y}\right)}{\partial y}=\frac{j\eta_{h}k_{h}}{h}\;$$

-momentum equations:


(8)
$$\rho_{e}\left(v_{x}\frac{\partial v_{x}}{\partial x}+v_{y}\frac{\partial v_{x}}{\partial y}-v_{\theta }{}^{2}\frac{R^{\prime }}{R}\right)=-\frac{\partial p}{\partial x}+\mu \frac{\partial^{2}v_{x}}{\partial y^{2}}\;$$



(9)
$$\rho_{e}\left(v_{x}\frac{\partial v_{\theta }}{\partial x}+v_{y}\frac{\partial v_{\theta }}{\partial y}+v_{x}v_{\theta }\frac{R^{\prime }}{R}\right)=\mu \frac{\partial^{2}v_{\theta }}{\partial y^{2}}\;$$


(10)$$0=-\frac{\partial p}{\partial y}\;$$
where: $v_{x},\;v_{\theta },\;v_{y}—$velocity components, p—pressure of the mixture of electrolyte and hydrogen, $\rho_{e}=\left(1-\beta \right)\rho_{e}^{0}—$electrolyte density, $\rho_{h}=\beta \rho_{h}^{0}$—hydrogen density, μ—dynamic coefficient of electrolyte viscosity.

Considering the Joule heat released due to the electric current flow, disregarding the mechanical energy dissipation, forced heat convection triggered by electrolyte flow, and heat exchange by the electrodes, the energy equation assumes the following form [8,10]:(11)$$v_{x}\frac{\partial T}{\partial x}+v_{y}\frac{\partial T}{\partial y}=\frac{\partial }{\partial y}\left(a\frac{\partial T}{\partial y}\;\right)+\frac{j_{A}^{2}}{\rho_{e}c_{p}\kappa }$$
where: T—temperature, a—thermal diffusivity, κ—electrolyte conductivity, $c_{p}—$the specific heat of the electrolyte. 

The system of Equations (6)–(11) formulated is a basic system of equations of flow for the analysis of the axisymmetric flow of the mixture of electrolyte and hydrogen in the IEG.

For the gap fed with electrolyte with a constant value of volumetric stream Q=const disregarding the local losses on the inlet and outlet, the solutions of Equations (6)–(10) must meet the boundary conditions for:-velocity components:
(12)$$\begin{array}{c} v_{x}=v_{y}=0\;\mathrm{for}\;y=0,\;v_{x}=0,\;\;\;v_{y}=0\;\mathrm{for}\;y=h\\ v_{\theta }=0\;\mathrm{for}\;y=0,v_{\theta }=\omega R\left(x\right)\;\mathrm{for}\;y=h \end{array}$$

-pressure:


(13)
$$p=p_{o}\;\mathrm{for}\;x=x_{o}$$


-for temperature:

(14)$$\begin{array}{c} \mathrm{on}\;\mathrm{the}\;\mathrm{walls}:\;T=T_{e}\;\mathrm{for}\;x\geq x_{i}\;\mathrm{and}\;y=0\;\mathrm{as}\;\mathrm{well}\;\mathrm{as}\;y=h\\ \mathrm{on}\;\mathrm{the}\;\mathrm{inlet}\;\mathrm{when}\;x=x_{i},\;T=T_{i} \end{array}$$
where: $p_{o}$—pressure on the IEG outlet, $x_{i},\;\;x_{o}$—IEG inlet and outlet coordinates, $T_{e}$—temperature of the electrodes, $T_{i}$—electrolyte temperature on the inlet.

From Equation (7), the distribution of volumetric concentration of the gas phase (the hydrogen released on the cathode) was received along the gap under the laminar flow conditions [34]:(15)$$\beta \left(x\right)=\frac{2\pi \eta_{H}k_{H}}{\mu_{H}Q}\frac{T\left(x\right)}{p\left(x\right)}\int_{x_{w}}^{x}j\left(x\right)Rdx$$
where: β—local average void fraction, $\rho_{Ho}=\frac{\mu_{H}\;p}{R_{H}T}$—hydrogen density, $\eta_{H}$—current efficiency of the hydrogen dissolution, $k_{H}$—electrochemical equivalent of hydrogen, $R_{H}$—hydrogen gas constant, $\mu_{H}$—molar mass of hydrogen, p,T—pressure and temperature.

The system of Equations (6)–(10) was analytically solved with the method of perturbation determining the velocity and pressure distributions in the IEG with a constant thickness [65].

Velocity and pressure distribution are presented in Appendix. The value of the gap thickness in a given cross-section is numerically determined by solving the evolution equation for a given time step. 

For a numerical solution of Equation (2), the Euler method was applied. It is a direct method, accurate enough in reference to the terms of the first order due to time step ∆t.

Time t has been presented on the time grid as a set of points:(16)$$t^{n}=t^{0}+n∆t\;$$
where: *n* = 0, 1, 2, 3…,*N*.

The TE and the WP were discretized in the global cylindrical coordinate system maintaining, respectively, for solids of revolution, the axisymmetric, namely assuming that:(17)$$R_{i}=R_{0}+i∆R\;$$
where: i=0, 1, 2, 3…I, $∆R=\frac{R_{o}-R_{i}}{I}$.

Replacing the derivatives which occur in Equation (2) with simple differential terms, index A of the selected point on the anode WP with index i, the following sequence of algebraic equations was obtained in a form of [8]:

when i=0:(18)$$Z_{i}^{n+1}=Z_{i}^{n}-\left[k_{V}j_{A}\sqrt{1+{\left(\frac{Z_{i}^{n}-Z_{i+1}^{n}}{∆R}\right)}^{2}}\right]∆t\;$$
when 0<i<I:(19)$$Z_{i}^{n+1}=Z_{i}^{n}-\left[k_{V}j_{A}\sqrt{1+{\left(\frac{Z_{i-1}^{n}-Z_{i+1}^{n}}{2∆R}\right)}^{2}}\right]∆t\;$$
when i=I:(20)$$Z_{i}^{n+1}=Z_{i}^{n}-\left[k_{V}j_{A}\sqrt{1+{\left(\frac{Z_{i-1}^{n}-Z_{i}^{n}}{∆R}\right)}^{2}}\right]∆t\;$$

Discrete differential Equations (18)–(20) allow, in successive time iterations ∆t, to determine new anode WP coordinates in the global Cartesian coordinate system Z,R (Figure 2).

To solve energy Equation (11), the finite difference method was applied, determining the temperature distribution across and along the gap. Considering the accuracy and stability of the numerical diagram for solving the problem, the Crank-Nicholson method was applied [8].

Replacing the derivatives which occur in Equation (11) with discrete differential terms in a form of:(21)$${\frac{\partial T}{\partial x}|}_{i,j}=\frac{T_{i,j}-T_{i-1,j}}{∆x_{i,j}}\;$$
(22)$${\frac{\partial T}{\partial y}|}_{i,j}=\frac{T_{i-1,j}-T_{i-1,j-1}}{∆y_{i}}\;$$
(23)$${\frac{\partial^{2}T}{\partial y^{2}}|}_{i,j}=\frac{T_{i,j+1}-2T_{i,j}+T_{i,j-1}}{2∆y_{i}^{2}}+\frac{T_{i-1,j+1}-2T_{i-1,j}+T_{i-1,j-1}}{2∆y_{i}^{2}}\;$$
(24)$${\frac{\partial a}{\partial y}|}_{i,j}=\frac{a_{i,j}-a_{i,j-1}}{∆y_{i}}\;$$
there was obtained a discrete form of Equation (11) from which one can calculate the value of temperature in successive grid nodes (Figure 3):
Figure 3Numerical grid of the electrolyte and hydrogen mixture flow area.
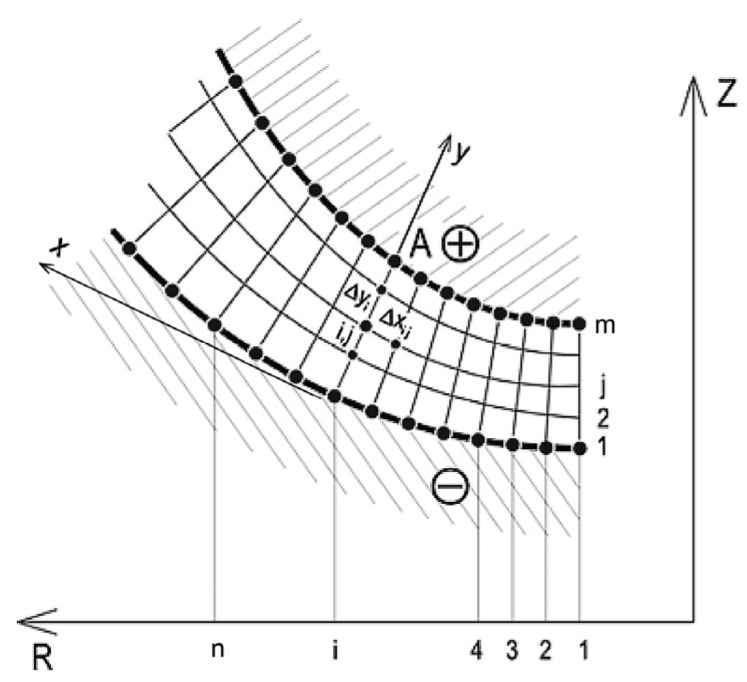

(25)$$\begin{array}{c} T_{i,j}^{k}=\frac{T_{i-1,j}+\gamma_{1}\left(T_{i-1,j+1}-2T_{i-1,j}+T_{i-1,j-1}+T_{i,j+1}^{k-1}+T_{i,j-1}\right)}{1+2\gamma_{1}}\;\\ +\frac{\gamma_{2}\left(T_{i-1,j}^{k-1}-T_{i-1,j-1}\right)-\frac{v_{y\;i,j}}{v_{x\;i,j}}\frac{∆x_{i,j}}{∆y_{i}}\left(T_{i-1,j}^{k-1}-T_{i-1,j-1}\right)+\gamma_{3}}{1+2\gamma_{1}}\; \end{array}$$
where: $\gamma_{1}=\frac{\left(a_{i,j}\right)∆x_{i,j}}{2v_{x\;i,j}∆y_{j}^{2}}$, $\gamma_{2}=\frac{\left(a_{i,j}\right)∆x_{i,j}}{v_{x\;i,j}∆y_{j}^{2}}$, $\gamma_{3}=\frac{j_{i}^{2}}{\rho_{i\;}c_{p}\varkappa_{i,j}\;v_{x\;i,j}}$.

index k—successive iteration number.

The coefficient of thermal diffusivity under the laminar flow conditions is determined with the formula:(26)$$a_{i,j}=\frac{\lambda }{\rho c_{p}}\;\;\lambda =\lambda_{0}\left(1-\beta \right)\;$$
where: λ—thermal conductivity, $\lambda_{0}$—thermal conductivity at $T_{0}$.

Defined with Formula (25), the approximate discrete differential equation was solved with the iteration method. Initial values$\;T_{i,j}^{k}$ were assumed and then, to calculate new values in successive iterations $T_{i,j}^{k}$, Formula (25) was applied. The iteration calculations are repeated to obtain the assumed calculation accuracy condition:(27)$$sup\left|T_{i,j}^{k}-T_{i,j}^{k-1}\right|\leq \varepsilon_{T}\;$$

The knowledge of physical electrolyte flow areas facilitated an effective determination of the WP (anode) shape.

### 2.3. Computer Simulation of ECM Process

The ECM simulation computer algorithm for the shape surfaces is presented in Figure 4.

### 2.4. Adaptive Control

The studies, the theory, and the experiments have demonstrated that, as for complex axisymmetric surfaces, in terms of machining accuracy, it is recommended to trigger the rotations of one of the electrodes [8,66]. At the same time, for such complex surfaces, it is very difficult to select the constant machining parameters valid for the input process time [67]. For that reason, adaptive control of the ECM process at the mathematical and numerical levels has been developed. The machining parameters have been assumed for which the simulation process will investigate the possibility of reaching the limit values (critical states), for which, in practice, the process is stopped. 

Should the critical states occur, system response algorithms were developed to correct the selected process parameters. The parameters correction is made in specific control points tc on the time axis occurring prior to the occurrence of the limit, critical state. The parameter correction is possible thanks to the stored ECM simulation results in points *tc*. Should the critical states occur, the machining simulation goes back in time to the control points identified earlier in which the adequate parameter modification is made [68].

As a result of such analysis, there are received variable-in-time ECM process parameters which facilitate avoiding the critical states. This is especially important with the ECM process for complex curved shapes. The final effect of such a simulation is the drill control code. The adaptive control diagram is presented in Figure 5.

The diagram specifies the machining parameters studied: the interelectrode gap thickness, electrolyte temperature, gas phase concentration, electrolyte flowrate, and the system responses. The parameters have been linked to the possible system response. The control code includes the following functions: N—block number, S—rotational speed of the electrode, F—electrode feed speed, G1—electrode infeed movement, and G4—electrode temporary stop. 

The level of critical values of adaptive control was adopted on the basis of the analysis of the simulation results of the ECM process. The critical values of the adaptive control parameters taken for the analysis are presented in Table 1 [54,63,69].

The scheme of adaptive control taking into account the kinematic input parameters is shown in Figure 6.

## 3. Results

### 3.1. Results of Simulation of ECM Process

Computer simulations based on the generalised solutions proper for any curvilinear rotary surfaces describing the fields of the electrolyte flow velocity and pressure in the interelectrode gap were made for a special surface, namely the hemispheric surface of revolution.

The geometric features of the WP and the TE are presented in Figure 7.

To enable computer simulation of ECM of rotary surfaces, an application in Delphi was developed based on the mathematical model of rotary surface ECM and the algorithm presented above.

For the calculations, the following key machining parameters have been considered (Table 2).

The calculations were carried out for three values of rotational speed WP (n = 0, 800, 1600 rpm) assuming a passivating electrolyte and constant machining time.

The calculation results $V_{x},\;p,\;T,\;j,\;\beta ,\;h$ are graphically presented in Figure 8, Figure 9, Figure 10, Figure 11, Figure 12 and Figure 13.

The application of anode rotation (WP) significantly affects the physical parameters of ECM of axisymmetric surfaces, especially the distribution along the IEG of mean velocity $v_{x}$, pressure p, and temperature T (Figure 8, Figure 9 and Figure 10). The non-linear distribution of these parameters is caused by the thickness of the inter-electrode gap changing during machining and the physical properties of the electrolyte (viscosity and conductivity). The occurring non-linearity is also caused by circumferential electrolyte flow velocity $v_{\theta }$, changing along the radius of the rotational surface, which causes, in particular at the gap outlet, clear differences $v_{x}$, p, and T. A non-linear pressure drop along the IEG can lead at significant rotational speeds and for excessively low electrolyte flow rates Q to dangerous negative pressure, in extreme cases allowing critical conditions to occur.

The application of a rotational movement of the WP slightly affects the hydrogen volume concentration distributions *β*, current density *j*, and local gap size h (Figure 11, Figure 12 and Figure 13). 

The volume of the gas phase (void fraction) increasing along the IEG, especially at the exit from the gap (Figure 12), results in a decrease in the treatment intensity. 

Changes in the local gap size along the IEG h caused by different electrochemical dissolution speeds, the curvature of the rotational surface, as well as changing physical conditions in the gap have a significant impact on the current density j distribution.

A comparison of these distributions suggests the so-called self-regulation phenomenon. This means that after some time the distribution of local gap size h, current density j, and other parameters set in time.

### 3.2. Experimental Verification

The experimental research carried out concerned the verification of two cases:mathematical model,adaptive control.

Experimental research was carried out on the test stand (Figure 14).

It was assumed that the machining process will be performed in the so-called machining cells, as a separate structural unit. Figure 15 presents the machining cell elements with a visible working electrode and a workpiece.

Detailed dimensions of the electrodes are shown in the drawing (Figure 7).

The machining cells were mounted in the machining part of the test stand. Figure 16 shows the machining part of the test stand.

Here one can specify the core made of the plates placed on the guides as well as the drive systems performing the process kinematics set. The core, in its central part, was fastened with a machining cell.

The simulation verifications of a mathematical model were made with an experiment with five samples (sample 1–sample 5). The shape of the sample (workpiece WP) is presented in Figure 7b. The ECM computer simulation results verification involved a comparison of the shape of the workpieces received from the calculations with the ECM results on the test stand. There were compared the generatrices of the surface received from a simulation and machining on the test stand. 

The ECM was made in the water solution of sodium nitrate (NaNO_3_) at a concentration of 15%. The selection of that electrolyte comes from a wide application of that agent in electrochemical machining. 

The profiles of working electrodes (ER), as well as the workpiece (PO), and the machining, were measured with the method point scanning with the coordinate measuring machine MISTRAL 070705 with the PH10M head (Figure 17).

To evaluate the accuracy of the mathematical electrochemical shape machining model applied, the following criteria were assumed:*δ*—distribution of the shape deviation (WP) calculated from the process computer simulation (theoretical simulation) and after the ECM,*δ_max_*—maximum deviation.

From shape deviation distribution *δ*, for each sample, there was a determined standard deviation *S* of shape deviation *δ* obtained from the equation:(28)$$S=\sqrt{\frac{\sum_{i=1}^{n}{\left(\delta_{i}-\bar{\delta }\right)}^{2}}{N-1}}\;$$
where: $\delta_{i}$—successive values of shape deviation, $\bar{\delta }$—arithmetic mean of the shape deviation for a given sample, N—number of elements in the sample (measuring points).

Figure 18 presents shape deviation distributions along coordinate *R* and the accuracy criteria assumed.

## 4. Discussion

It can be noted easily that the biggest deviations occurred always at the end of the gap (SM). It is a result of a high amount of gas and the ECM process products accumulating there, which can be accounted for with a considerable active machined surface. An increase in rotation speed results in a similar effect of error deviations establishing.

Adaptive control was verified for samples with the same geometrical and material characteristics as in the case of the mathematical model verification. 

Figure 19 presents the photograph of a sample with a spherical surface machined.

During ECM process studies with constant in-time machining parameter values, the so-called critical limit states were identified. The most frequent ones included: cavitation (Figure 20a), single short circuit (Figure 20b), and surface wave structure (Figure 20c).

Figure 21 shows a photograph of the surface machined with an adaptive control process.

The tests verifying the mechanism of the ECM adaptive control were performed for the surface of the sample with the geometry and the system of dimensions presented in Figure 5. Similarly, as for ECM simulation verification tests without applying the adaptive control mechanism, to evaluate the accuracy of the adaptive control mechanism proposed, there was an assumed distribution of set shape deviation δ (WP) and the shape of the workpiece (WP) produced after machining as well as standard deviation S of shape deviation δ. The measurement was made for two cross-sections of the sample with the scanning method by determining the shape deviation distribution along four generatrices of the surface (Figure 22).

The tests were repeated three times. The results are presented in Table 3.

Shape deviations for a specific sample measured along four generatrices do not deviate from one another in both cases of machining. A selection of variable machining parameters resulted not only in an increase in the process stability but also in shape and dimension repeatability of the surfaces produced.

## 5. Conclusions

The contemporary design of ECM technology for machine parts and tools, due to the cybernetic nature of the process, requires the use of computer machining simulations. Computer simulations can be implemented based on the analysis and mathematical modelling of the ECM process.

The work presented is an attempt at modelling the electrochemical machining process of curvilinear surfaces of revolution.

The theoretical analysis of ECM included determining the shape of the WP after a given machining time. To determine the shape of the WP, the equation of anode shape evolution known in theory was used. The equation of shape evolution was solved by the method of successive approximations, using the so-called time steps.

The analysis of the electrolyte mixture flow was based on the formulated generalized quasi 3D theoretical model of the ECM process of axisymmetric surfaces. Using the basic principles of behaviour: the principle of mass, momentum, and energy conservation for the multiphase medium (electrolyte, gas, electrochemical dissolution products), a system of equations was formulated governing the movement of the medium in the IEG.

To solve the equation resulting from the principle of energy conservation, the finite difference method was used.

The analysis facilitates the developing complex simulation algorithms which, at the same time, analyse and modify the ECM process for any axisymmetric surfaces. The analysis, in terms of critical states, has covered the distribution of the IEG, temperature, gas phase concentration, and the electrolyte flow rate. 

The causes of the occurrence of critical states and, as a result, limit states are most frequently related to inadequately selected machining conditions. It is mostly due to a change in the medium properties, especially a decrease in electrical conductivity, resulting from the products of the ECM process accumulated in the inter-electrode gap and release of gases, mostly hydrogen. It inevitably leads to short circuits and process instability. Another important factor is a variable electrolyte flow in the interelectrode gap, in the range: 2300 < Re < 50,000.

The described process of mathematical modeling and adaptive control was verified experimentally by determining the distribution of shape deviations *δ*, maximum shape deviation *δ_max_*, and mean standard deviation *S*. The obtained results confirm the usefulness of the proposed method. The method of ECM process modelling and controlling has enhanced the machining stability and accuracy, especially for the electrodes with complex axisymmetric shapes.

## Figures and Tables

**Figure 1 materials-15-07751-f001:**
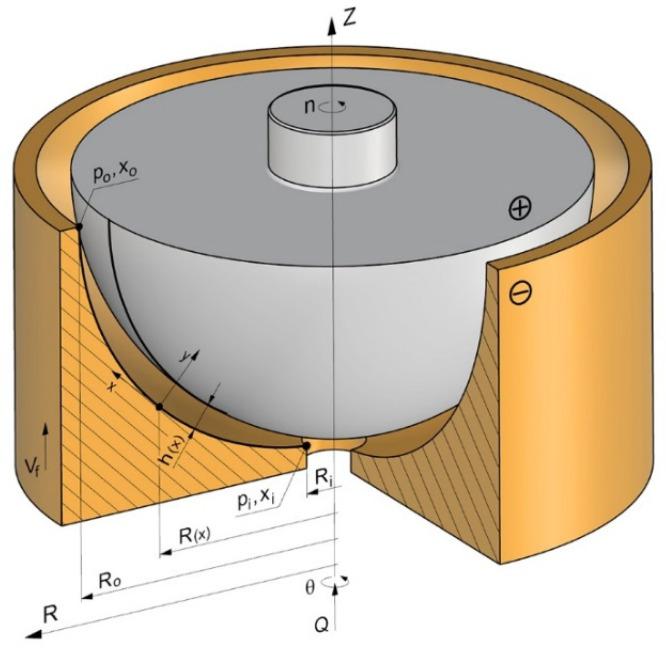
Area flow of electrolyte in IEG. *x_i_*, *R_i_*, *p_i_*—inlet coordinate, inlet pressure, *x_o_*, *R_o_*, *p_o_*—outlet coordinate, outlet pressure, h(x)—interelectrode gap, *V_f_*—feed of the TE, Q—flow rate, *n*—rotation speed of the WP.

**Figure 2 materials-15-07751-f002:**
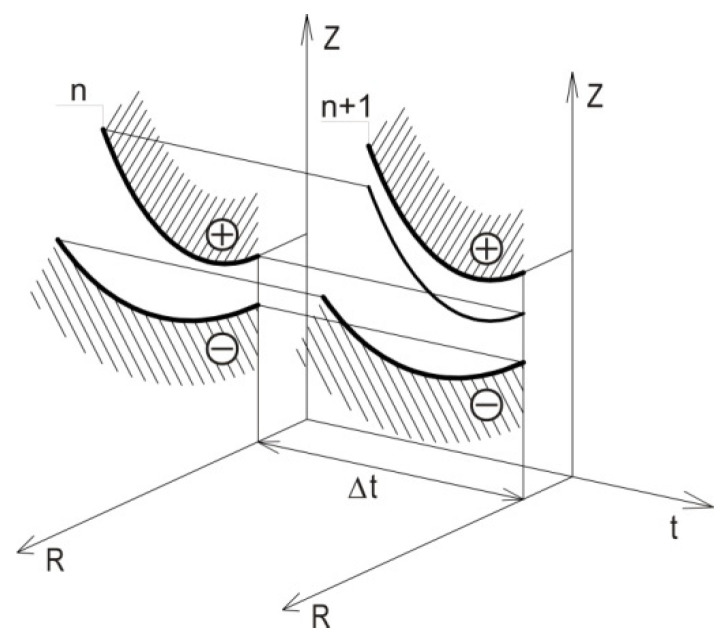
Effect of ECM time on shape evolution (WP).

**Figure 4 materials-15-07751-f004:**
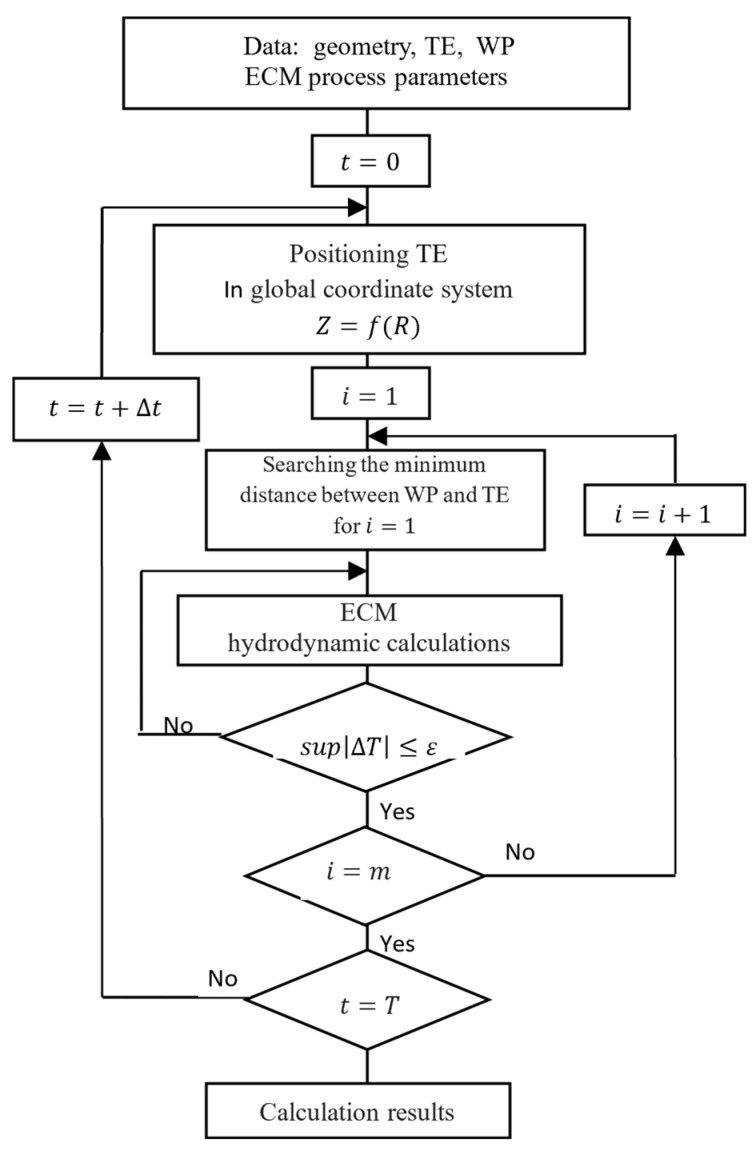
ECM process simulation algorithm (markings: *i*, *m*—current, and the last point of the curve describing the WP).

**Figure 5 materials-15-07751-f005:**
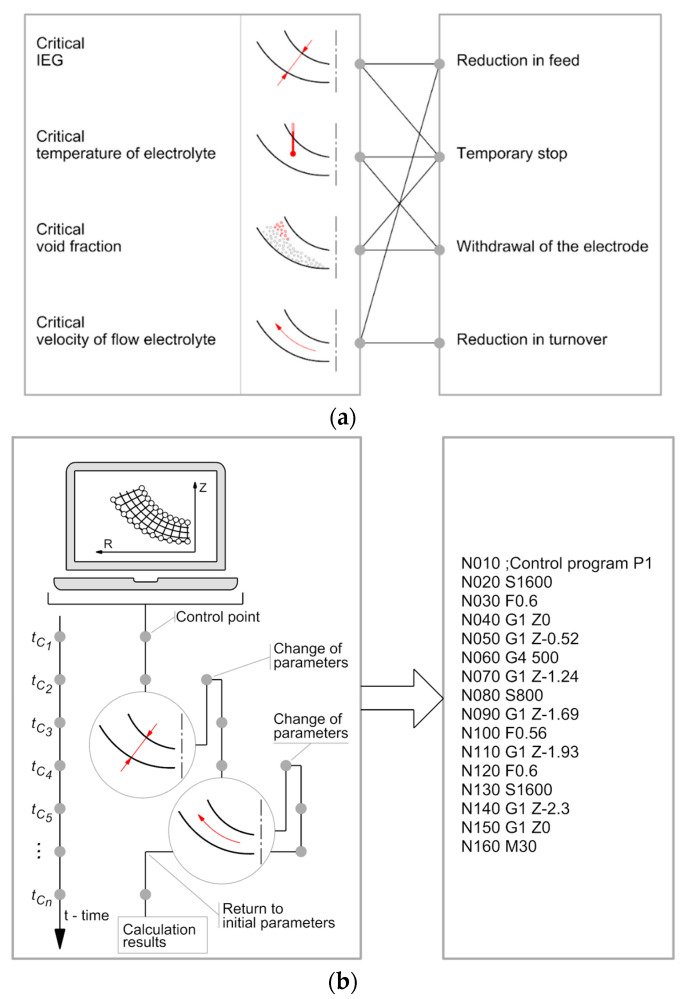
Adaptive control diagram: (**a**) the critical states and device response identification, (**b**) machining simulation and control program.

**Figure 6 materials-15-07751-f006:**
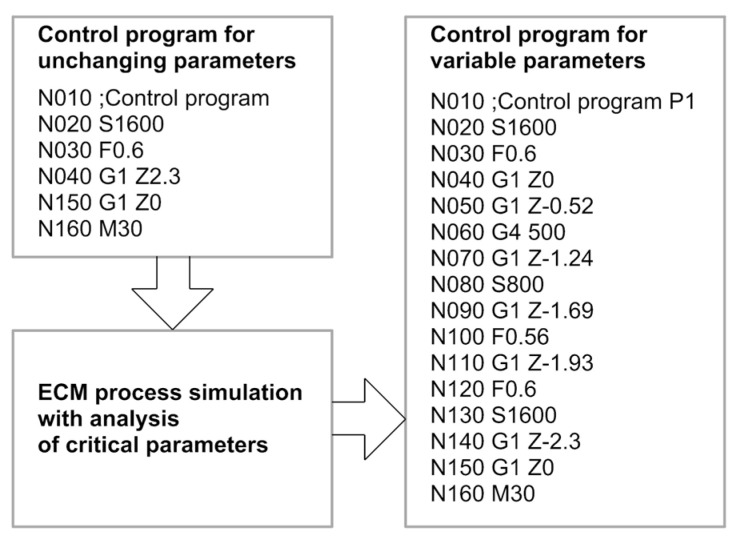
Adaptive control diagram.

**Figure 7 materials-15-07751-f007:**
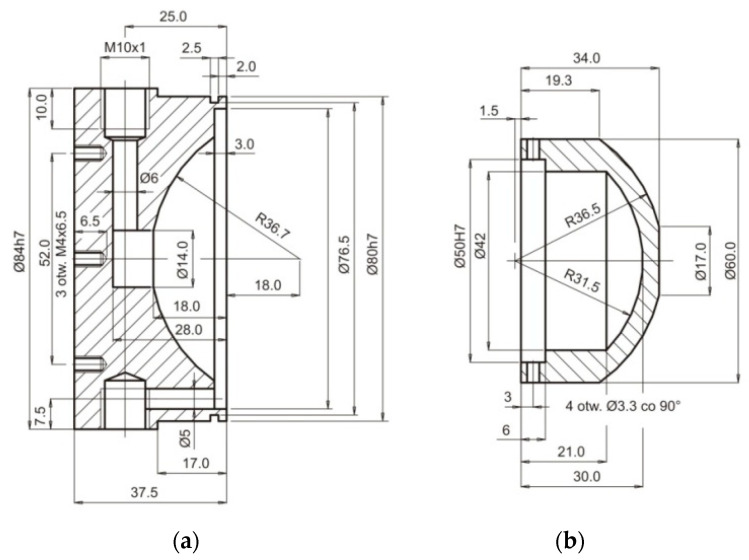
Geometric features: (**a**) TE—material—electrolytic copper, (**b**) WP—material—hot work steel 2312.

**Figure 8 materials-15-07751-f008:**
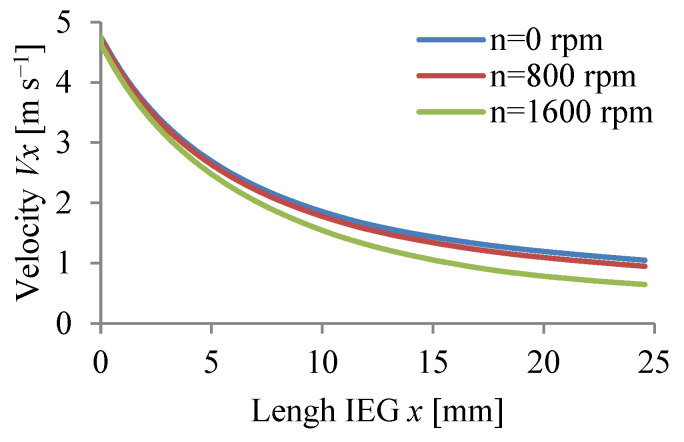
Velocity distribution *v_x_*.

**Figure 9 materials-15-07751-f009:**
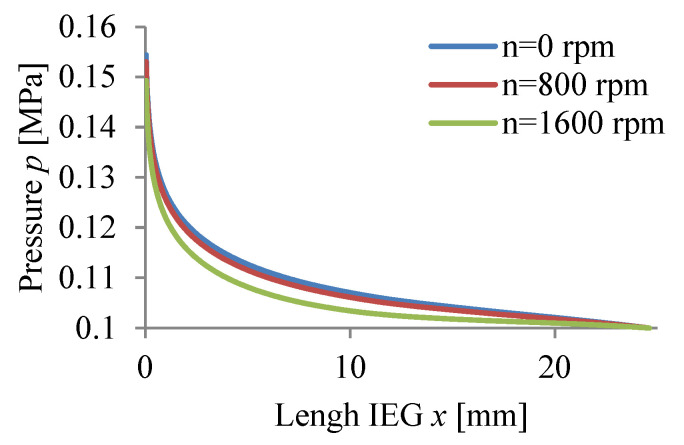
Pressure distribution *p*.

**Figure 10 materials-15-07751-f010:**
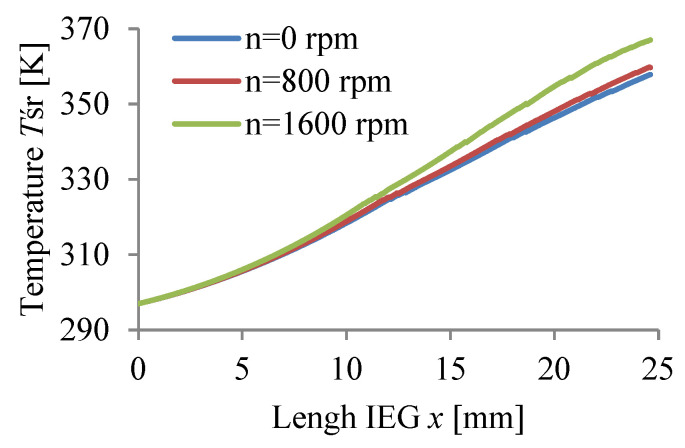
Temperature distribution *T*.

**Figure 11 materials-15-07751-f011:**
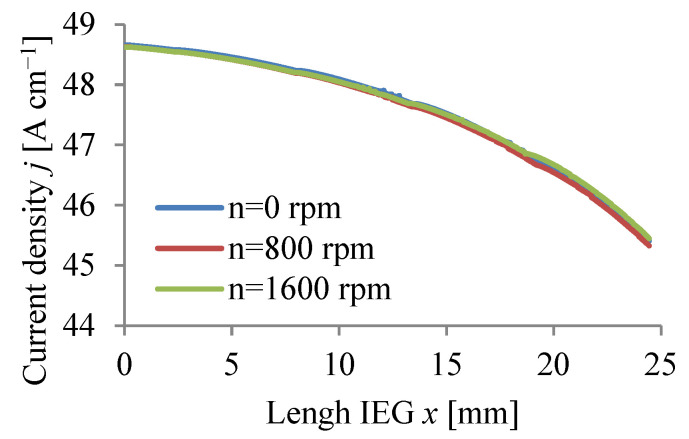
Current density distribution *j*.

**Figure 12 materials-15-07751-f012:**
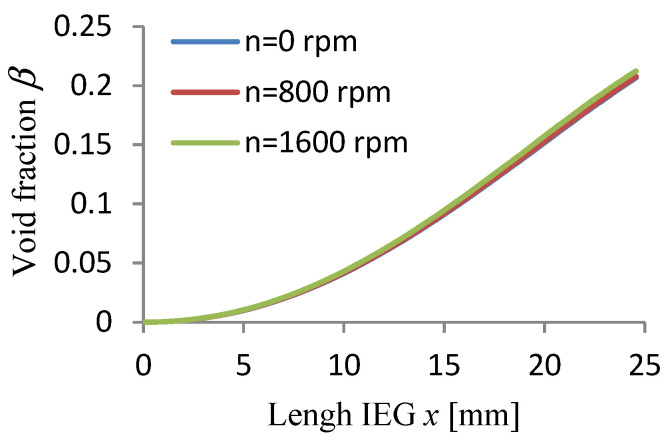
Void fraction distribution *β*.

**Figure 13 materials-15-07751-f013:**
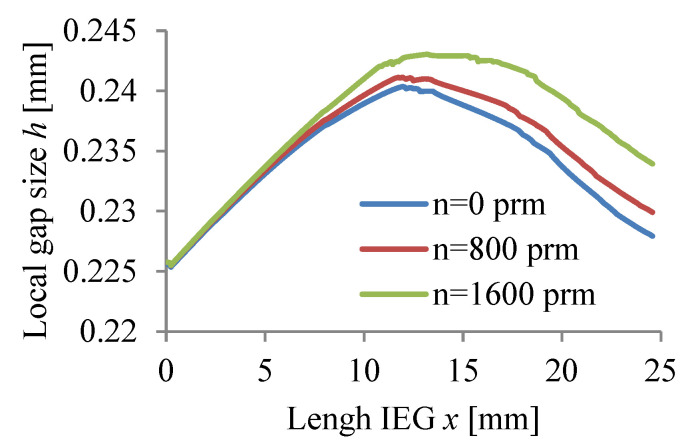
Local gap size distribution *h*.

**Figure 14 materials-15-07751-f014:**
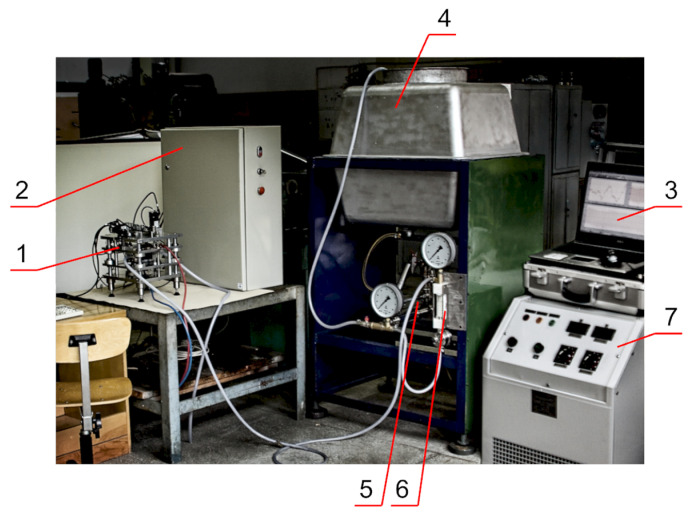
Test stand: processing part 1, the control system 2, computer 3, electrolyte tank 4, hydraulic pump 5, flow regulator 6, DC power supply 7.

**Figure 15 materials-15-07751-f015:**
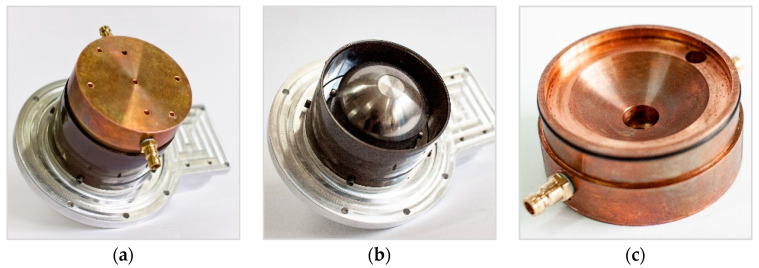
Machining cell: (**a**) prepared to work, (**b**) workpiece, (**c**) working electrode.

**Figure 16 materials-15-07751-f016:**
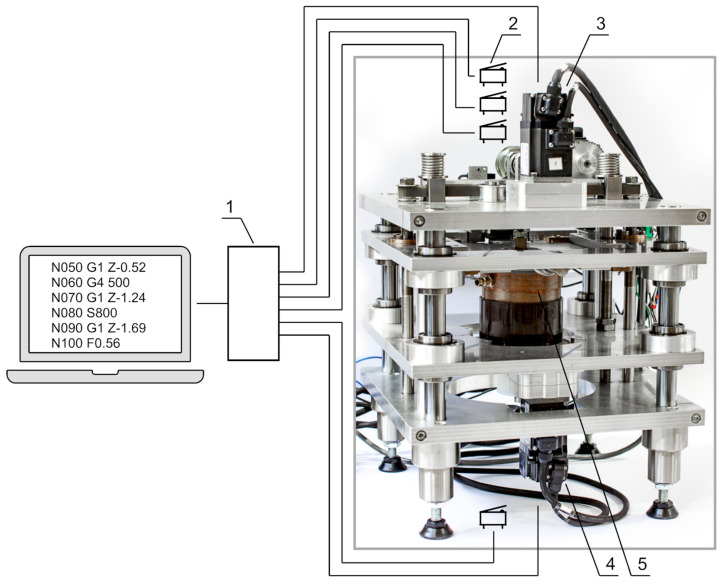
Elements of the control system: 1—Mitsubishi FX 3U driver with an FX 3U-232-BD I/O module, 2—limit switches, 3—DC motors (servo)—feed motion *V_f_*, 4—DC motors (servo)—rotary movement, 5—machining cells.

**Figure 17 materials-15-07751-f017:**
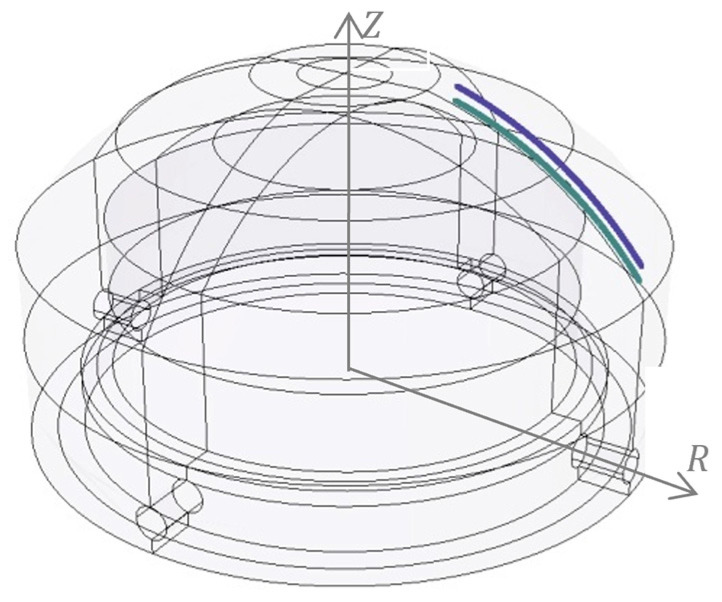
System of generatrices for the shape deviation distribution analysis.

**Figure 18 materials-15-07751-f018:**
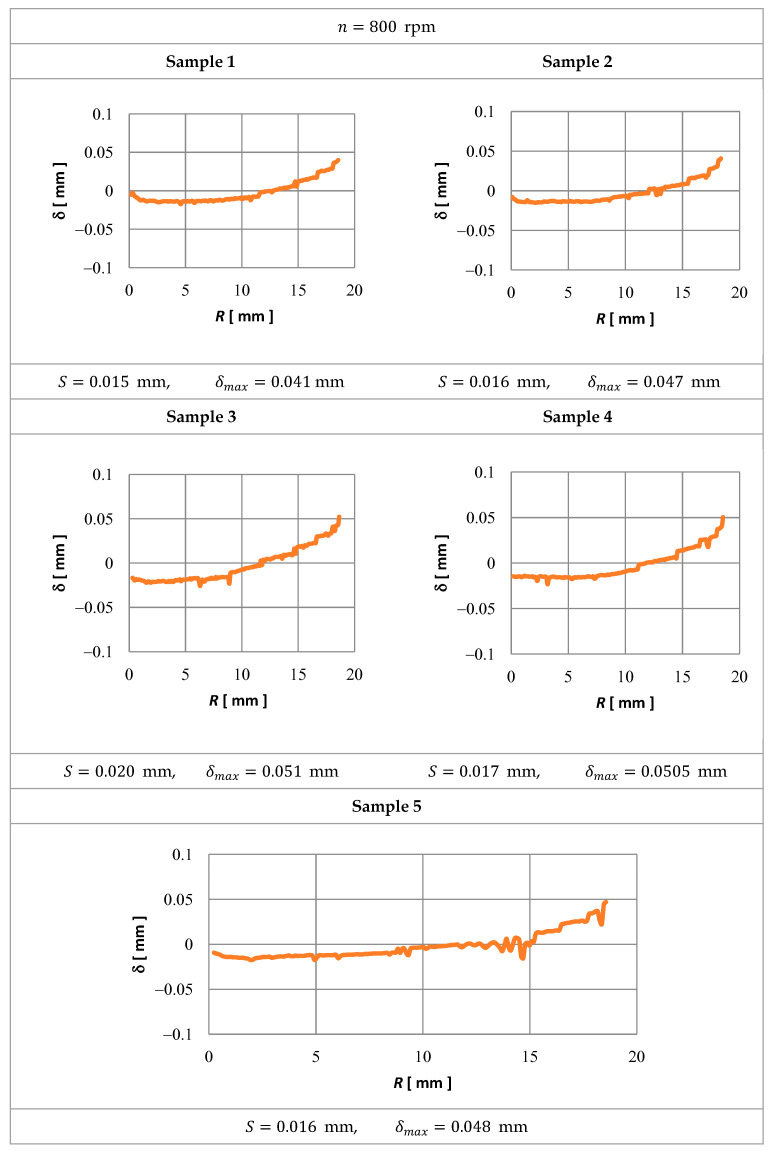
Distribution of shape deviation δ along the length of coordinate R.

**Figure 19 materials-15-07751-f019:**
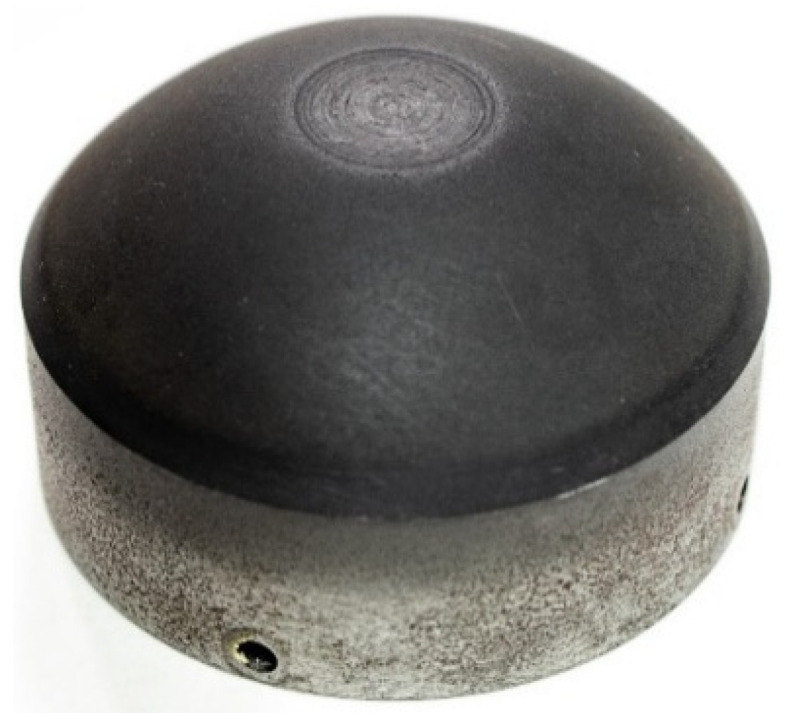
Sample machined.

**Figure 20 materials-15-07751-f020:**
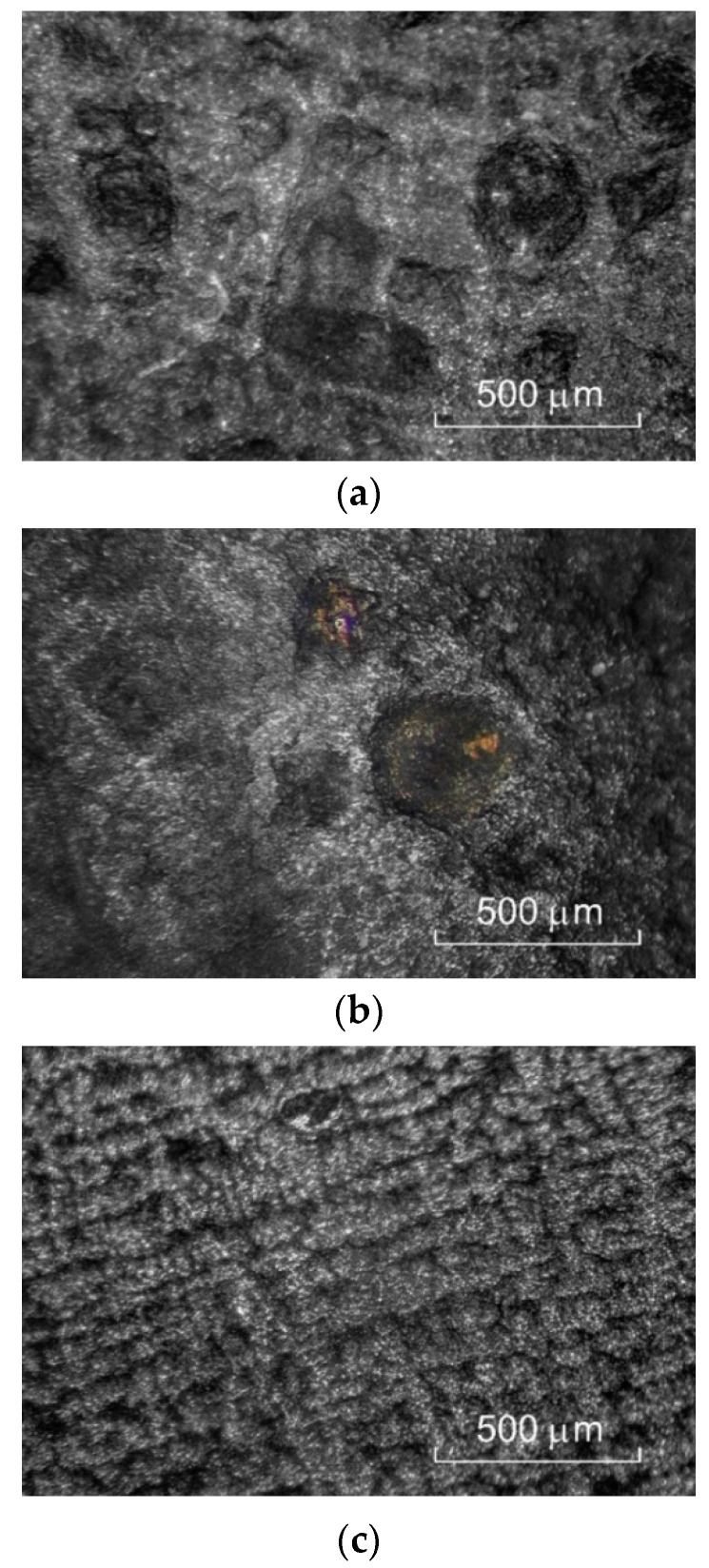
View of the sample surface once the critical machining states occurred: (**a**) cavitation, (**b**) single short circuit, (**c**) wave structure.

**Figure 21 materials-15-07751-f021:**
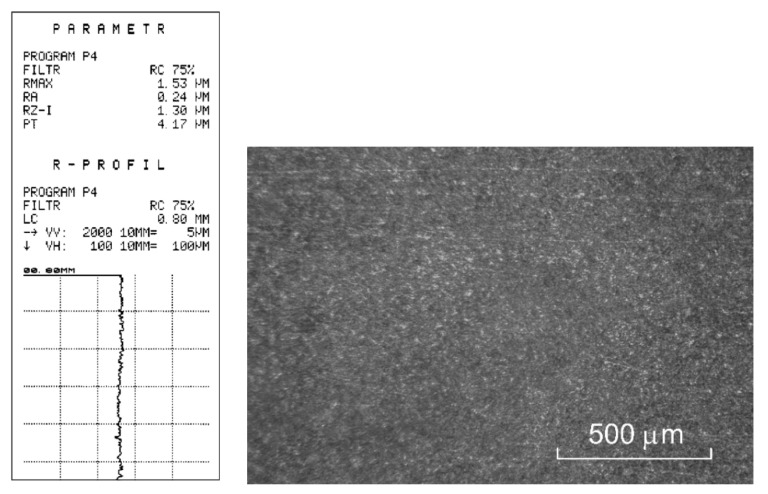
View of the sample surface adequately machined.

**Figure 22 materials-15-07751-f022:**
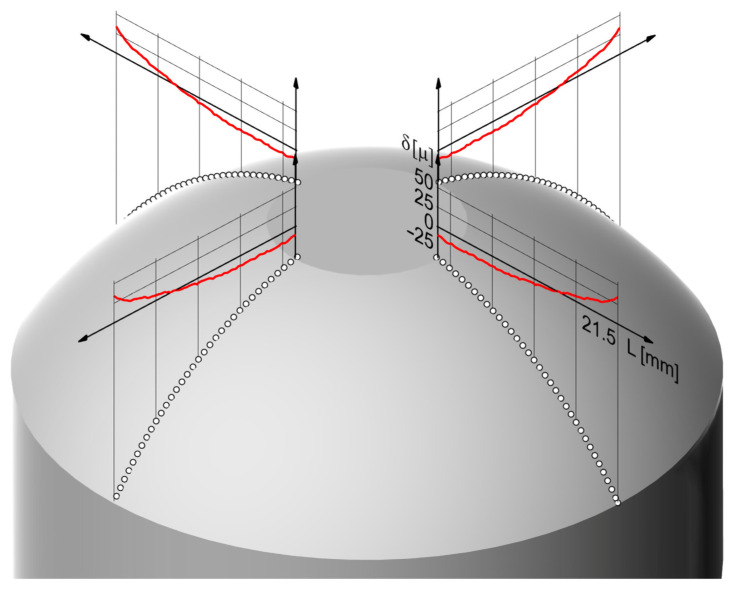
Distributions of shape deviation δ recorded in the studies with an adaptive control mechanism (δ_max_ = 0.033 mm, S = 0.012 mm).

**Table 1 materials-15-07751-t001:** Critical values of adaptive control parameters.

Parameters	Specifications
Critical IEG	0.05 mm
Critical temperature of electrolyte	368 K
Critical void fraction	0.45
Critical velocity of flow electrolyte	5 ms^−1^

**Table 2 materials-15-07751-t002:** Machining parameters.

Parameters		Specifications
Initial gap	h	0.2 mm
Feed rate of the TE	$V_{f}$	1 mm min^−1^
Working voltage	U	15 V
Volumetric flow rate	Q	3 dm^3^ min^−1^
Outlet pressure	$p_{o}$	0.1 MPa
Machining time	t	60 s
Workpiece material	WP	alloy tool steel 2312
Electrochemical machinability:	$k_{V}$	=1.59 (1 − exp(2.56 − 0.112*j*) mm^3^ (A min)^−1^, *j* A cm^−2^
electrolyte		15% water solution of NaNO_3_

**Table 3 materials-15-07751-t003:** Results of verification tests for a case with an adaptive control mechanism.

Sample No.	*δ* [mm]	*S* [mm]
Sample 1	0.33	0.012
Sample 2	0.34	0.013
Sample 3	0.33	0.012

## Data Availability

The data presented in this publication will be made available on reasonable request.

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
