# Peer review of "Electrochemical Machining of Curvilinear Surfaces of Revolution: Analysis, Modelling, and Process Control"

_materials, 2022, doi:10.3390/ma15217751_

Round 1
Reviewer 1 Report
Simulation-based design or development of the ECM process is the research frontiers in this field. On the other hand, this task is difficult due to the complexity of ECM process, especially when the workpieces feature complicated geometry.
The authors provide a novel model and its solution for the adaptive control of ECM of the axisymmetric components of any curvilinear shape. And its computation accuracy seems good according to the experiments. Thus, this model is potential for effective develop the process. The contents are arranged logically.
In my opinion, this paper could be accepted in present form. Only few misspellings should be corrected, eg. EMC process.
Author Response
Thank you for your positive review of the article. We have removed the noticed spelling mistakes.
Reviewer 2 Report
1. There are 41 references in the manuscript that have been published for more than 20 years, It is recommended to cite studies in recent years in the introduction section for comparison.
2. The order of references is not in accordance with the order of text citations. It is recommended to rearrange the references.
3. When verifying the results of the mathematical model simulation, the authors compared the deviation from the standard workpiece size. How is the dimensional deviation from the real ECM? The author mentions this in the manuscript, but does not describe it. The relationship between these three dimensional deviations should be described.
4. When verifying the adaptive ECM, the entire adaptive process was not introduced in the manuscript, what adjustments have been made to the parameters?
5. The manuscript focuses on the modeling of the electrochemical machining process of the rotating surface. However, the workpiece rotational speed does not appear in the process of establishing the mathematical model. What is the mechanism by which the workpiece rotational speed affects the machining process? Does it affect the machining gap distribution or the state distribution in the gap?
Author Response
- There are 41 references in the manuscript that have been published for more than 20 years, It is recommended to cite studies in recent years in the introduction section for comparison.
The greatest development of ECM machining took place at the end of the 20th century. In the attached list of references, we tried to include those items that relate to the ECM process modeling and works related to the analysis of problems occurring in the ECM process. We tried to describe the works of these authors in the introduction to the article.
- The order of references is not in accordance with the order of text citations. It is recommended to rearrange the references.
The citations of the articles included in the reference list have been improved. Thank you for this remark.
- When verifying the results of the mathematical model simulation, the authors compared the deviation from the standard workpiece size. How is the dimensional deviation from the real ECM? The author mentions this in the manuscript, but does not describe it. The relationship between these three dimensional deviations should be described.
To assess the accuracy of the applied mathematical model of the ECM machining, the workpiece shape deviation distribution calculated on the basis of computer simulation of machining and the workpiece shape deviation distribution obtained after machining were adopted. On this basis, the standard deviation was determined. The standard deviation is a very frequently used measure that characterizes the dispersion well, therefore this type of measure and the maximum deviation were used to assess the accuracy.
- When verifying the adaptive ECM, the entire adaptive process was not introduced in the manuscript, what adjustments have been made to the parameters?
The article in Figures 5 and 6 (on the right side) contains all the code that controls the ECM process generated by the adaptive control system. The called new parameters are: time standstill, new rotational speed ER, new feed ER, return to the set feed, return to the set rotation speed. Changing the parameters allowed to avoid critical states in the ECM.
- The manuscript focuses on the modeling of the electrochemical machining process of the rotating surface. However, the workpiece rotational speed does not appear in the process of establishing the mathematical model. What is the mechanism by which the workpiece rotational speed affects the machining process? Does it affect the machining gap distribution or the state distribution in the gap?
The rotational speed of the workpiece, and precisely the angular velocity of the workpiece, appears in the boundary conditions concerning the momentum equation related to the circumferential movement of the electrolyte in the inter-electrode gap. Solving this equation leads to the determination of the speed distribution taking into account the rotational speed of the workpiece. In order to determine the radial velocity, the peripheral speed is necessary, hence this distribution also includes this (rotational) speed.
In the following steps, the speed distributions are used to solve the energy equation. The obtained temperature distributions must therefore take into account the rotational speed of the workpiece entered into the speed distributions.
On the basis of the already known temperature distributions in the inter-electrode gap, the equation for the evolution of the shape of the workpiece was solved.
Reviewer 3 Report
The Manuscript "Electrochemical machining of curvilinear surfaces of revolution: analysis, modelling and process control" has been reviewed, and it was observed that authors put efforts in preparing the manuscript. It is interesting and useful for the readers. However, some minor modification are required.
1. The abstract section should be supported by some statistical results.
2. Authors used many citations at one place. It needs to break. max. 2-4 citation van be placed at one place. eg. 26........66 in last line of introduction section.
3. How the process parameters are controlled in the present modelling.
4. How the electrolyte mixture flow has controlled during the ECM process.
5. In the conclusion section, it also needs to update with some statistical data.
Author Response
1. The abstract section should be supported by some statistical results.
The abstract section has been revised to include research information.
2. Authors used many citations at one place. It needs to break. max. 2-4 citation van be placed at one place. eg. 26........66 in last line of introduction section.
Bearing in mind the above remark, we grouped the cited works anew, trying to limit the citation in a given sentence to a few items from the literature list.
3. How the process parameters are controlled in the present modelling.
Parameters in modeling the ECM process are controlled due to four critical states: critical IEG, critical temperature of electrolyte, critical void fraction, critical velocity of electrolyte flow.
4. How the electrolyte mixture flow has controlled during the ECM process.
The cell supply system consists of a hydraulic unit equipped with a constant flow rate regulator of the electrolyte mixture. The use of a flow regulator allowed to obtain a constant value of the volumetric flow rate of the electrolyte, regardless of the load occurring in the inter-electrode gap. Pressure sensors are built into the hydraulic supply system of the machining cell to record the pressure upstream and downstream of the cell. The process parameters control system is based on the Mitsubishi FX 3U controller coupled with a PC.
5. In the conclusion section, it also needs to update with some statistical data.
The conclusion was revised to include research information.
Reviewer 4 Report
There are several questions.
1. In table 2, the tool electrode feedrate is 0.1mm/s, that means 6mm/min. Please confirm.
2. It presented the simulation results with different rotation speed. Which equations reflect the effect?
3. Did the authors consider the differences of the original WP sample as well as its influences on surface profile, because it may not reach the equilibrium status in 60 seconds?
4. The references could be deleted and replace some more relative literature. For example, http://dx.doi.org/10.1016/j.jmatprotec.2013.07.012.
Author Response
1. In table 2, the tool electrode feedrate is 0.1mm/s, that means 6mm/min. Please confirm.
The feed speed of the electrode is 1 mm / min. The value of this speed given in table 2 is incorrect.
We corrected this parameter in the article.
2. It presented the simulation results with different rotation speed. Which equations reflect the effect?
The rotational speed of the workpiece, and precisely the angular velocity of the workpiece, appears in the boundary conditions concerning the momentum equation related to the circumferential movement of the electrolyte in the inter-electrode gap. Solving this equation leads to the determination of the speed distribution taking into account the rotational speed of the workpiece. In order to determine the radial velocity, the peripheral speed is necessary, hence this distribution also includes this (rotational) speed.
In the following steps, the speed distributions are used to solve the energy equation. The obtained temperature distributions must therefore take into account the rotational speed of the workpiece entered into the speed distributions.
On the basis of the already known temperature distributions in the inter-electrode gap, the equation for the evolution of the shape of the workpiece was solved.
3. Did the authors consider the differences of the original WP sample as well as its influences on surface profile, because it may not reach the equilibrium status in 60 seconds?
The shape of the workpiece was pre-shaped on a lathe. As a result of preliminary simulation calculations, it was possible to determine the machining time after which the equilibrium state was achieved.
4. The references could be deleted and replace some more relative literature. For example, http://dx.doi.org/10.1016/j.jmatprotec.2013.07.012.
Thank you for the suggestion of extending the reference list. We included the proposed article in our literature list.